# Detecting Proximal Caries on Periapical Radiographs Using Convolutional Neural Networks with Different Training Strategies on Small Datasets

**DOI:** 10.3390/diagnostics12051047

**Published:** 2022-04-21

**Authors:** Xiujiao Lin, Dengwei Hong, Dong Zhang, Mingyi Huang, Hao Yu

**Affiliations:** 1Fujian Provincial Engineering Research Center of Oral Biomaterial, School and Hospital of Stomatology, Fujian Medical University, Fuzhou 350005, China; cherishlin686@gmail.com (X.L.); denthdw@163.com (D.H.); 2Department of Prosthodontics, School and Hospital of Stomatology, Fujian Medical University, Fuzhou 350005, China; 3College of Computer and Data Science, Fuzhou University, Fuzhou 350025, China; zhangdong@fzu.edu.cn (D.Z.); hmy9029@163.com (M.H.); 4Department of Applied Prosthodontics, Graduate School of Biomedical Sciences, Nagasaki University, Nagasaki 852-8521, Japan

**Keywords:** neural networks, proximal caries, training strategy, small dataset, periapical radiograph

## Abstract

The present study aimed to evaluate the performance of convolutional neural networks (CNNs) that were trained with small datasets using different strategies in the detection of proximal caries at different levels of severity on periapical radiographs. Small datasets containing 800 periapical radiographs were randomly categorized into a training and validation dataset (*n* = 600) and a test dataset (*n* = 200). A pretrained Cifar-10Net CNN was used in the present study. Different training strategies were used to train the CNN model independently; these strategies were defined as image recognition (IR), edge extraction (EE), and image segmentation (IS). Different metrics, such as sensitivity and area under the receiver operating characteristic curve (AUC), for the trained CNN and human observers were analysed to evaluate the performance in detecting proximal caries. IR, EE, and IS recognition modes and human eyes achieved AUCs of 0.805, 0.860, 0.549, and 0.767, respectively, with the EE recognition mode having the highest values (*p* all < 0.05). The EE recognition mode was significantly more sensitive in detecting both enamel and dentin caries than human eyes (*p* all < 0.05). The CNN trained with the EE strategy, the best performer in the present study, showed potential utility in detecting proximal caries on periapical radiographs when using small datasets.

## 1. Introduction

Globally, dental caries is the most common oral disease, with 2.3 billion people suffering from caries of permanent teeth and more than 530 million children suffering from caries of deciduous teeth [1]. In China, an increasing caries prevalence is observed in line with the fourth national oral health epidemiological survey, with results demonstrating a prevalence of 38.5% in permanent teeth and 71.9% in deciduous teeth, respectively [2,3,4]. Dental caries occurs when plaque-associated bacteria produce acid that demineralizes the tooth. Controlling oral microbial biofilms is crucial for preventing dental caries. However, dental caries develops despite the use of antibiotics since bacterial resistance occurs due to excessive antibiotic use [5]. Generally, tooth loss is mainly attributed to dental caries [6], which is related to detrimental dietary changes and may lead to gastrointestinal disorders, even increasing the risk of Alzheimer’s disease [7,8]. To manage dental caries, especially early caries lesions, precise detection is required before non-invasive or invasive treatment [9,10]. In particular, initial caries lesions occurring on the proximal surface in premolars and molars usually require auxiliary examination [11] since initial proximal caries lesions are difficult to detect by clinical examination unless the disease is advanced [12].

Intraoral radiographs, including bitewing radiographs and periapical radiographs, are commonly used to assist the diagnosis of proximal caries [13,14]. Akarslan et al. [15] compared the diagnostic accuracy of bitewing radiographs, periapical radiographs, and panoramic radiographs for proximal caries detection in posterior teeth. Both bitewing and periapical radiographs demonstrated a mean area under the receiver operating characteristic curve (AUC) that was higher than 0.9, indicating excellent performance. However, the performance of bitewing radiographs in detecting early caries lesions was somewhat contradictory, as they have been reported to have higher sensitivity than periapical radiographs [16] and a low diagnostic yield [17]. According to previous studies, only approximately 60% of proximal caries lesions were detected on bitewing radiographs [18,19]. Notably, bitewing radiographs are limited in their ability to offer information that allows cavitated and non-cavitated lesions to be distinguished from one another in the initial stages of progression [20]. In terms of periapical radiographs, a systematic review including 117 studies revealed that a low sensitivity of 42% was found for the detection of proximal caries [21]. Regrettably, a noteworthy limitation of periapical radiographs is that 40% of the tooth tissue has been demineralized when caries is successfully diagnosed by human eyes [12,22]. Thus, seeking a method to improve the diagnostic accuracy of dental caries on intraoral radiographs is of great significance.

Recently, convolutional neural networks (CNNs), a class of deep learning algorithms, have been widely applied in dentistry [23,24]. For example, CNNs have been applied to evaluate dental caries in bitewing and periapical radiographs [9] and periodontal bone loss in periapical or panoramic radiographs [25]. Lee et al. [9] explored the performance of CNNs in detecting dental caries lesions in periapical radiographs, obtaining an accuracy of 82.0% with a dataset of 3000 periapical images. According to a recent review, at least 1000 CT training datasets were required to obtain 98.0% validation accuracy with deep learning; also, 4092 CT training datasets were required to reach the desired accuracy of 99.5% [24]. CNNs are far more data hungry due to the millions of learnable parameters that they estimate [23]. Collecting data and making ground truth labels are essential to establish a successful deep learning project since these labels are used to train and test a model [23]. However, acquiring high-quality labelled data can be costly and time-consuming [23]. Notably, it is difficult to secure a large medical dataset due to patient privacy and security policies [26]. Therefore, strategies to improve the accuracy of CNNs trained with small datasets should be explored [27].

In general, the procedure used to carry out the learning process is called the training strategy; this strategy is applied to the neural network to obtain the best possible loss and increase accuracy [28]. In previous studies, different training strategies, such as different preprocessing strategies (e.g., contrast enhancement and average subtraction) and data augmentation were conducted to improve the performance of CNNs [29]. GoogLeNet achieved the best performance (96.69% accuracy) with the original images, while AlexNet performed better (94.33% accuracy) by using average subtraction [29]. Interestingly, Khojasteh et al. [30] introduced a novel layer in CNNs in which a preprocessing layer (e.g., contrast enhancement) was embedded followed by the first convolutional layer; this approach increased the accuracy of CNNs from 81.4% to 87.6%. Different strategies may work for different networks. Based on the current evidence, it should be considered that if small datasets (fewer than 1000 units per group [24]) of periapical radiographs were obtained, different training strategies, such as image preprocessing before training, could be adopted to improve diagnostic accuracy [27,29,31]. However, limited studies have focused on the recognition differences in neural networks with different training strategies (e.g., different preprocessing strategies) in dentistry, especially using small datasets. In addition, information regarding the performance in detecting dental caries at different levels of severity (different levels of caries progression) is scarce. Therefore, the present study aimed to evaluate the performance of a deep learning-based CNN in detecting proximal caries at different levels of progression on periapical radiographs, in which the CNN was trained with small datasets using different strategies. The following null hypotheses were tested: (1) no differences would be found in the performance of the trained CNN; and (2) the trained CNN would be more sufficient and accurate than human eyes in detecting proximal caries.

## 2. Materials and Methods

The research was performed following the principles of the Declaration of Helsinki and received approval from the Research Ethics Committee at the School and Hospital of Stomatology, Fujian Medical University (approval no.: 2018Y0029; approval date: 20 June 2018). The current study followed the guidelines of the Standards for Reporting of Diagnostics Accuracy Studies (STARD).

### 2.1. Study Design

In the present study, in which the CNN was trained with small datasets using different strategies, the performances of human observers and a deep learning-based CNN in evaluating proximal caries at different levels of severity on periapical radiographs were compared.

In this study, a pretrained Cifar-10Net CNN network was used as a classification model to distinguish caries from non-caries. Cifar-10Net was applied for its better efficiency object recognition [32]. Cifar-10Net is the basic network model used to classify the Cifar-10 dataset and is frequently used in image recognition [32]. As a subset of the larger dataset of 80 million tiny images, Cifar-10 included 60,000 colour images that contained 10 object classes [33].

According to previous study, different metrics were deployed to assess the classification performance of human observers and the CNN, including the diagnostic accuracy, sensitivity, specificity, positive predictive value (PPV), negative predictive value (NPV), receiver operating characteristic (ROC) curve, AUC, a precision-recall (P-R) curve, and the F1-score (F1-score = 2 × precision × recall/(precision + recall)).

### 2.2. Reference Dataset

Anonymous periapical radiographs were collected from patients who visited the Hospital of Stomatology, Fujian Medical University, from 2019 to 2020, following the randomization principle. All the periapical radiographs were taken by radiologists applying the paralleling technique [34]. Periapical radiographs obtained from the patient archiving and communication system (PACS) (Infinitt PACS, Infinitt Healthcare Co. Ltd., Seoul, Korea) were downloaded and saved in a bitmap image (BMP) file format [9]. The metadata, e.g., age, sex, and image creation date, were also obtained. Periapical radiographs with proximal caries limited to the crown or integral proximal surface were selected, excluding those with restorations and with severe noise, haziness, distortion, and shadows [9]. Periapical radiographs were cropped into images containing two posterior teeth to meet the training requirements; for inclusion, one tooth suffered from proximal caries (caries occurred in 1 or 2 proximal surfaces) and the other tooth was intact. All images were clearly revalidated, and proximal caries (including enamel and dentin caries of permanent teeth) were distinguished from non-proximal caries by 3 endodontists independently. No clinical records were acquired or evaluated in the procedure [35]. The 3 examiners all had more than 5 years of clinical experience [35]. For a single image, a consensus of the 3 examiners was required to identify the dental caries. Discussion was carried out when inconsistent evaluations arose. Periapical radiographs were excluded when disputes remained unsolved. To reduce the diagnostic bias which that might be caused by image cropping, original periapical radiographs were also provided to 3 the examiners for further needs. Consequently, small datasets of 800 periapical radiographs matching the training requirements were generated from 3165 periapical radiographs. The included radiographs were from 385 men and 415 women (mean age: 45.3 years). All 800 periapical radiographs were given a random number by using the RAND function and were randomly assigned to the training or test dataset by using the data sorting function in Microsoft Excel (Microsoft office 2016, Microsoft, Redmond, WA, USA). Subsequently, a training and validation dataset (*n* = 600) and a test dataset (*n* = 200) were randomly generated. Original datasets were then converted to grayscale images using uniform parameters, which was called the normalization of images.

### 2.3. Data Processing

#### 2.3.1. Image Preprocessing

The training dataset of 600 periapical radiographs was preprocessed in MATLAB (MATLAB 2016b, MathWorks, Natick, MA, USA). Three preprocessing strategies of image recognition (without image preprocessing, IR), image segmentation (IS) [36] and edge extraction (EE) [31] were employed; IR and EE images were then overlaid into the original periapical radiographs. IS was performed based on a marked watershed segmentation algorithm [36]. The Canny operator was used when the image was preprocessed by means of EE [31]. An alpha transparency blending algorithm was utilized in the process of image superposition.

#### 2.3.2. Image Labelled in MATLAB

The training dataset of 600 periapical radiographs was uploaded to the app in MATLAB used to label the images; caries lesions were marked using a training image labeler (TIL) based on the agreement among the 3 examiners and shown as the region of interest (ROI). According to the ROI, the level of caries severity was then evaluated. Caries progression was evaluated based on the following criteria [37]: level 0, non-proximal caries; level 1, proximal caries limited to the outer half of the enamel; level 2, proximal caries limited to the inner half of the enamel; level 3, proximal caries limited to the dento–enamel junction (DEJ); level 4, proximal caries limited to the outer half of the dentin; and level 5, proximal caries limited to the inner half of the dentin.

### 2.4. Training the CNN

A pretrained Cifar-10Net CNN network was used in the present study, which consists of an input layer, convolutional layer, rectified linear unit (ReLU) layer, pooling layer, fully connected layer, SoftMax layer, and output layer [33,38]. The convolutional, ReLU, and pooling layer form the core building blocks of the CNN. Specifically, the convolutional layer was responsible for updating filter weights during the data training; the ReLU layer mapped image pixels to the semantic content of the image; the pooling layer down sampled the data flowing through the network [33]. Before the output layer, the SoftMax layer, which acted as a classifier [39], received a two-dimensional vector from the fully connected layer and subsequently decided on the caries. Transfer learning was used to train the data to prevent overfitting, in which some parameters of the pretrained Cifar-10Net CNN network were transferred to the targeted Cifar-10Net CNN network [40]. Taking the loss value as the evaluation metric, a base learning rate of 0.0001 was set, and 400 epochs were run. Fine-tuning was conducted during transfer learning to improve diagnostic accuracy [9]. No standardized grayscale thresholding was used in the present CNN because the Cifar-10Net CNN is a nonlinear network instead of a regressor that needs a threshold [33,41]. Different training strategies implementing IR, IS, and EE were used to train the CNN independently [29], consequently generating three kinds of training models.

### 2.5. Test

The test process was carried out on the recognition model using a test dataset with no labels. Different recognition modes were established based on the training models, which were correspondingly distinguished as IR, IS, and EE. Finally, the detection of dental caries was conducted through the CNN algorithm that was trained, in which original images were pre-processed with IR, IS, EE and then analysed. Image superposition was performed between the original and preprocessed images when IS and EE strategies were used to detect proximal caries lesions. The diagnostic process of different recognition modes is shown in Figure 1. In addition, the workflow process of the CNN is exhibited in Figure 2.

The main functions of relevant codes and some parameters in Data processing and Training were conducted as follows: IS, function imgf = fenge(rgb); EE, function imgCanny = edge_canny(I,gaussDim,sigma,percentOfPixelsNotEdges,thresholdRatio); Image superposition, function C = diejia(pic_1,pic_2); Training, function training = trainRCNNObjectDetector (Unnamed, mylayers, options, … ‘NegativeOverlapRange’, [0 0.3]).

### 2.6. Human Observers

Proximal caries on original periapical radiographs from the test dataset with no label was also assessed by the other 3 endodontists who had 3 to 10 years of clinical experience [35]. These images served as a comparator group that was used to gauge the performance of different recognition modes against that of human eyes [35]. Consensus should be achieved among 3 human observers when diagnosing proximal caries.

The test dataset was evaluated according to the evaluation criteria mentioned above and was used as the gold standard to compare the performance of IR, IS, EE and human observers.

### 2.7. Statistical Analysis

For different recognition modes and human eyes, the metrics to evaluate the performances were compared using the chi-square test and Z test. The *p* value was set at 0.05, and the 95% confidence interval (CI) was assessed.

## 3. Results

Consistency among examiners was checked before the revalidation, and Kendall’s W coefficient of 0.830 (*p* < 0.001) showed strong consistency. The caries occurrences in proximal surfaces in the reference dataset, that is, the evaluation from the three examiners, are depicted in Table 1. The diagnostic accuracy, sensitivity, specificity, PPV, and NPV, including the 95% CI, for the detection of proximal caries using different recognition modes and human eyes are shown in Table 2.

A comparison of the ROC curves and P-R curves are shown in Figure 3 and Figure 4, respectively, for both different recognition modes and human eyes. For the IR recognition mode, the AUC was 0.805 (95% CI 0.771~0.838). In the case of the EE recognition mode, the AUC was 0.860 (95% CI 0.832~0.888). Regarding the IS recognition mode, the AUC was 0.549 (95% CI 0.508~0.589). In the case of human eyes, the AUC was 0.767 (95% CI 0.732~0.802). The AUCs of IR and EE recognition modes were both significantly greater than that of the IS recognition mode (*p* all < 0.001). The AUC of the EE recognition mode was significantly higher than that of the IR recognition mode (*p* = 0.013). Compared to human eyes, only the AUC of the EE recognition mode was significantly higher (*p* < 0.001). The IR, EE, and IS recognition modes and human eyes achieved F1-scores of 0.766, 0.837, 0.292 and 0.724, respectively.

ROC = receiver operating characteristic; IR = image recognition; EE = edge extraction; IS = image segmentation.

P-R = precision-recall; IR = image recognition; EE = edge extraction; IS = image segmentation.

A comparison of the performance of IR, EE, and IS recognition modes and human eyes in detecting proximal caries at different levels of severity is demonstrated in Table 3. A comparison of the performance of IR, EE, and IS recognition modes and human eyes in detecting proximal caries at the enamel and dentin levels is exhibited in Table 4.

## 4. Discussion

Based on the present findings, the null hypotheses that no differences would be found in the performance of the trained CNN and that the trained CNN would be more sufficient and accurate than human eyes in the detection of proximal caries were partially accepted. In particular, the CNN trained with EE and IR strategies performed better than that with the IS strategy; and the CNN trained with the EE strategy achieved higher accuracy and sensitivity than human eyes in the detection of proximal caries.

Early intervention can remineralize softened enamel, which can block or reverse the process of dental caries [42]. Thus, finding an approach to detect initial caries, especially proximal caries, efficiently is of great significance [35]. Various diagnostic technologies have been developed to overcome the limitations of clinical and radiographic diagnosis and to improve the accuracy of caries detection [9]. Deep-learning-based CNNs are a class of artificial neural networks that are attracting interest across various fields, including radiology [23]. Compared to natural images, medical images are thought to have unique characteristics and are well fitted to deep learning [26]. Recently, using deep learning to detect dental caries lesions on periapical radiographs [9] and bitewing radiographs [35] has been studied. Compared to human eyes, deep-learning-based CNNs showed a satisfying discerning ability in detecting dental caries on periapical radiographs or bitewings [35]. A recent study revealed that approximately half of proximal caries lesions that reached the outer half of dentin were cavitated [43]. Moreover, it was suggested that restorations should be restricted to cavitated lesions [20], advising infiltration and sealing to manage non-cavitated proximal lesions as well as proximal lesions limited to one third of the outer dentin [20,44]. Thus, distinguishing proximal caries into different levels of severity is important for its guiding significance in dental treatment [43]. However, the performance of CNNs in detecting proximal caries at different levels of severity has not yet been reported. Notably, considering the difficulty in obtaining massive amounts of labelled medical data and patient security and confidentiality, different training strategies, such as image preprocessing, were conducted to search for the solution [23,26,27,29]. Pertinently, the present study first trained the CNN with small datasets (fewer than 1000 units per group) using different training strategies; that is, the CNN was trained with IR, IS, and EE strategies, which correspondingly resulted in different trained neural networks. Importantly, periapical radiographs were selected because of their clinical usage [45]. The AUCs of different trained neural networks (referred to as different recognition modes) were ultimately calculated and compared because of their significance in diagnostic performance [46].

Mandrekar et al. [46] suggested that an AUC of 0.8 to 0.9 is considered excellent in diagnostic performance. Accordingly, the recognition modes of IR and EE performed exceptionally in detecting proximal caries in the present study, with AUCs of 0.805 and 0.860, respectively. However, the IS recognition mode, with an AUC of 0.549, showed no discrimination in detecting proximal caries [46]. The EE recognition mode showed the greatest accuracy and achieved significantly higher accuracy than the human eye; thus, it was proposed that the EE recognition mode should be considered for small datasets. Edges are produced by the transition between various areas in the image, which is one of the most basic feature signals in the image signal [31]. For periapical radiographs, the changes in greyscale partly produce image edges. Pertinently, image edge extraction plays an important role in image recognition and processing [31]. In the present study, CNNs performed better than those in a previous study, which may be due to the Canny operator used in [31]. The use of the Canny operator strengthened the edge feature, which enabled CNNs to detect edges more efficiently [31]. The poor performance of the IS recognition mode may be attributed to excessive extrema and noise [36]. Furthermore, Lee et al. [9] reported that an AUC of 0.845 was achieved on both premolar and molar models based on a CNN. The EE recognition mode achieved an AUC of 0.860. Given the suggestions from Lee et al. [9], fine-tuning and transfer learning technology were used in the present study, which may account for the minor differences.

Based on the present findings, the highest sensitivity was obtained by the EE recognition mode, which might be due to the use of the Canny operator [31]. More specifically, the EE recognition mode was more sensitive than the IS recognition mode for the detection of enamel and dentin caries. However, the EE recognition mode did not demonstrate its superiority until level 4 caries detection (proximal caries limited to the outer half of dentin) and level 5 detection (proximal caries limited to the inner half of dentin) compared with the IR recognition mode. This contradictory phenomenon may result from the limited sample size. The EE, IR, and IS recognition modes all showed satisfying specificity, which could not be ignored when high sensitivity was achieved [9].

Based on the present results, the EE recognition mode was significantly more sensitive than human eyes for the detection of enamel and dentin caries. In terms of level 1 (proximal caries limited to the outer half of enamel), the EE recognition mode achieved higher sensitivity than the human eye, which was consistent with a previous report that enamel caries on periapical radiographs could be detected by human observers only after caries lesions advanced into the outer half of enamel [9]. It was probable that the EE recognition mode combined the greyscale changes and the features of caries edges, making this approach more capable of detecting initial caries even when learning on small datasets [31].

In addition, the P-R curve was observed, and the F1-score was calculated to assess the performance of the recognition model in cases where the dataset was unbalanced (e.g., the number of caries samples and non-caries samples differed extremely in quantity) [47]. The P-R curve was established by plotting data with precision (PPV) on the *y*-axis and recall (sensitivity) on the *x*-axis [48]. The F1-score was the harmonic of the precision and recall, which represents agreement with truth [48]. In the present study, the EE recognition mode achieved a precision score of 0.808, a recall score of 0.869, and the highest F1-score of 0.837. These scores were higher than those reported in the study performed by Srivastava et al. [48], which achieved a precision score of 0.615, a recall score of 0.805, and an F1-score of 0.700 for the detection of tooth caries in bitewing radiographs using deep learning. The high recall of our recognition model showed that the model missed only a few proximal caries from the ground teeth [48]. More importantly, the high precision indicated that few false positives occurred based on the high sensitivity [48].

According to previous studies, GoogLeNet and U-Net were used to detect dental caries on periapical radiographs and bitewings, respectively [9,35]. Compared to the approaches utilizing GoogLeNet and U-Net, different models equipped with different loss functions and combinations of the parameters contributed to the main differences among existing approaches and the present proposal [9,35]. GoogLeNet, using a dataset of 3000 periapical radiographs, showed a sensitivity of 81.0%, a specificity of 83.0%, a PPV of 82.7% and an NPV of 81.4% [9]. U-Net, utilizing a dataset of 3686 bitewings, obtained a sensitivity of 75.0%, a specificity of 83.0%, a PPV of 70.0% and an NPV of 86.0% [35]. Notably, the present EE results were a sensitivity of 86.9%, a specificity of 85.2%, a PPV of 80.8% and an NPV of 90.0% based on the small dataset of 800 periapical radiographs; thus, a better performance was found in the present CNN compared to that of GoogLeNet. Differences between the present CNN and U-Net may be due to the different training strategies and detection objects [49]. Therefore, a preprocessing strategy that is common but well suited to medical images, such as EE strategy, was proposed to preprocess periapical radiographs that are commonly used in clinical practice [45].

Fine-tuning, one way to utilize a pretrained network [23], was the method selected to pretrain Cifar-10Net in the present study. Transfer learning was applied since this approach allowed generic features learned on a sufficiently large dataset to be shared with seemly disparate datasets [23]. Compared to other networks, such as AlexNet [50], Cifar-10Net has fewer layers and faster recognition speed, which partly reduces the recognition rate.

Several limitations should be considered in the present study. First, radiological dosage standardization was lacking since changes in the applied radiological dosage may occur for individual oral conditions, for example, soft tissue conditions [51]. Establishing a record of the applied dosage when taking periapical radiographs could be considered for obtaining standard images, which should be accomplished in cooperation with radiologists. Second, in the absence of a “hard” reference test, only radiographical evaluations were conducted, lacking clinical evaluations [9,35]. Furthermore, inconsistent with the CNN trained with EE and IR strategies that performed well, the CNN trained with IS strategy, namely, the IS recognition mode, behaved indiscriminately in detecting proximal caries and showed a poorer performance than that achieved by human eyes. Moreover, the sample size was unbalanced at different levels of caries severity, which may have impacted the present findings. A larger sample size and balanced dataset (different levels of caries severity) could be considered to enhance the generalizability of the present approach and exploring the impact of using the network on treatment decisions. Additionally, the recognition rate was partly sacrificed to increase the training and recognition speed [38]. The number of network layers should be increased to improve the recognition rate in future studies. Last, a further clinical comparison group (such as combining the clinical records when caries evaluations are conducted) to indicate the false-positive and false-negative rates of the calibrated examiners and the Cifar-10Net CNN process could be considered to further verify the current findings.

## 5. Conclusions

Within the limitations of a lack of standardization of the radiological dosage and a small sample size, we prudently concluded that the deep-learning-based CNN trained with the EE strategy performed excellently in detecting proximal caries on periapical radiographs; different training strategies, such as image preprocessing, could be considered to improve the accuracy of the CNN model, especially when a small dataset was used. Pertinently, the present proposed method should be regarded as a computer-aided caries detection system in clinical practice, in which clinical evaluations should be combined and not discarded. However, the challenges of how the proposed method could be generalized and applied to treatment decisions should be considered. Additionally, regarding the limitation of only conducting the radiographical evaluations, a further clinical comparison group to indicate the false-positive and false-negative rates of the calibrated examiners and the Cifar-10Net CNN process could be considered to further verify the current findings.

## Figures and Tables

**Figure 1 diagnostics-12-01047-f001:**
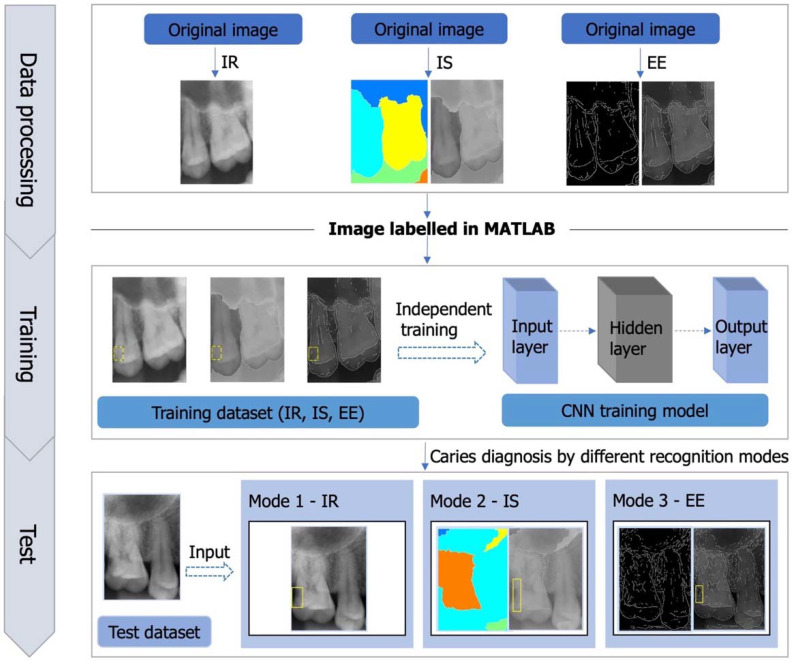
Proximal caries detection on periapical radiographs using deep learning with different recognition modes.

**Figure 2 diagnostics-12-01047-f002:**
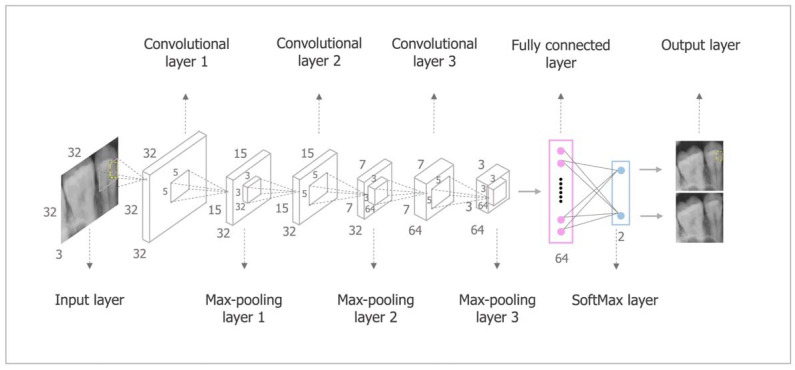
The workflow process of the CNN.

**Figure 3 diagnostics-12-01047-f003:**
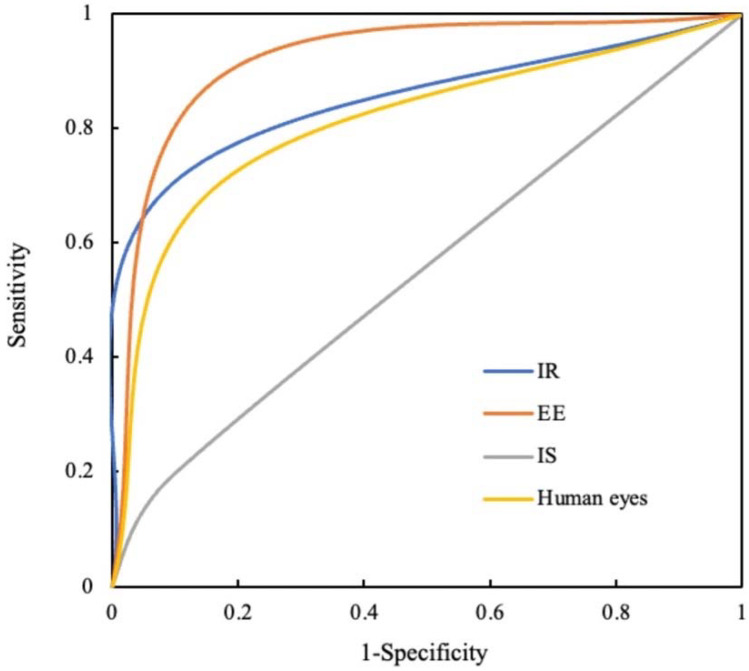
The ROC curves of different recognition modes and human eyes.

**Figure 4 diagnostics-12-01047-f004:**
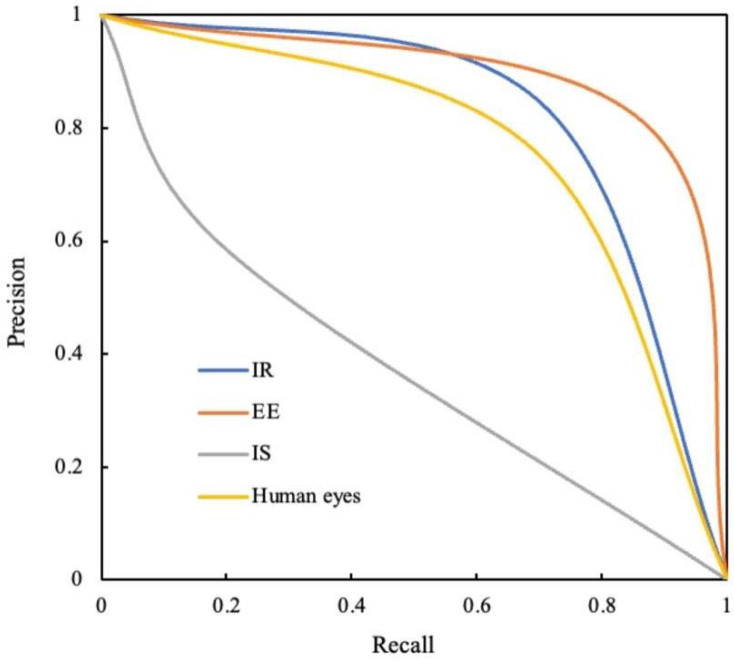
The P-R curves of different recognition modes and human eyes.

**Table 1 diagnostics-12-01047-t001:** Caries occurrences in proximal surfaces in the reference dataset.

Dataset	Level 0	Level 1	Level 2	Level 3	Level 4	Level 5
Training dataset	1289	53	139	78	336	505
Test dataset	465	15	55	35	83	147
Overall	1754	68	194	113	419	652

**Table 2 diagnostics-12-01047-t002:** Accuracy, sensitivity, specificity, PPV, and NPV for the detection of proximal caries using different recognition modes and human eyes.

Recognition Mode	Accuracy (%, 95% CI)	Sensitivity (%, 95% CI)	Specificity (%, 95% CI)	PPV (%, 95% CI)	NPV (%, 95% CI)
IR	82.1 (79.5~84.8) ^a,b^	70.1 (65.2~75.1) ^a^	90.8 (88.1~93.4) ^a^	84.5 (80.3~88.8) ^a^	80.8 (77.5~84.2) ^a^
EE	85.9 (83.5~88.3) ^a^	86.9 (83.2~90.5) ^b^	85.2 (81.9~88.4) ^a,b^	80.8 (76.8~84.9) ^a^	90.0 (87.2~92.8) ^b^
IS	60.6 (57.2~64.0) ^c^	19.4 (15.2~23.7) ^c^	90.3 (87.6~93.0) ^a^	59.1 (49.8~68.4) ^b^	60.9 (57.2~64.5) ^c^
Human eyes	78.0 (75.1~80.1) ^b^	69.0 (64.0~73.9) ^a^	84.5 (81.2~87.8) ^b^	76.2 (71.4~81.1) ^a^	79.1 (75.5~82.7) ^a^

Different lowercase letters in a column indicate significant differences in different recognition modes and in human eyes.

**Table 3 diagnostics-12-01047-t003:** A comparison of the performance of IR, EE, and IS recognition modes and human eyes in detecting proximal caries at different levels of severity.

Recognition Mode	Level 0 (Sample, %)	Level 1 (Sample, %)	Level 2 (Sample, %)	Level 3 (Sample, %)	Level 4 (Sample, %)	Level 5 (Sample, %)
IR	422/465 (90.8%) ^a^	8/15 (53.3%) ^a,b^	33/55 (60.0%) ^a,b^	18/35 (51.4%) ^a^	49/83 (59.0%) ^a^	127/147 (86.4%) ^a^
EE	396/465 (85.2%) ^a,b^	10/15 (66.7%) ^a^	42/55 (76.4%) ^a^	28/35 (80.0%) ^a^	70/83 (84.3%) ^b^	141/147 (95.9%) ^b^
IS	420/465 (90.3%) ^a^	3/15 (20.0%) ^a,b^	8/55 (14.5%) ^c^	5/35 (14.3%) ^b^	14/83 (16.9%) ^c^	35/147 (23.8%) ^c^
Human eyes	393/465 (84.5%) ^b^	2/15 (13.3%) ^b^	28/55 (50.9%) ^b^	19/35 (54.3%) ^a^	53/83 (63.9%) ^a^	129/147 (87.8%) ^a,b^

Different lowercase letters in a column indicate significant differences in different recognition modes and in human eyes.

**Table 4 diagnostics-12-01047-t004:** A comparison of the performance of IR, EE, and IS recognition modes and human eyes in detecting proximal caries at the enamel and dentin levels.

Recognition Mode	Enamel (Sample, %)	Dentin (Sample, %)
IR	41/70 (58.6%) ^a,b^	194/265 (73.2%) ^a^
EE	52/70 (74.3%) ^a^	239/265 (90.2%) ^b^
IS	11/70 (15.7%) ^c^	54/265 (20.4%) ^c^
Human eyes	30/70 (42.9%) ^b^	201/265 (75.8%) ^a^

Different lowercase letters in a column indicate significant differences in different recognition modes and in human eyes.

## Data Availability

The data presented in this study are available on request from the corresponding author. The data are not publicly available because of privacy restrictions.

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
