# Peer review of "Detecting Proximal Caries on Periapical Radiographs Using Convolutional Neural Networks with Different Training Strategies on Small Datasets"

_diagnostics, 2022, doi:10.3390/diagnostics12051047_

Round 1
Reviewer 1 Report
TITLE: Detecting proximal caries on periapical radiographs using convolutional neural networks with different training strategies on small datasets
Diagnostic MDPI- 1671220
The aim of the present investigation was to assess the performance of convolutional neural networks (CNNs) that were trained with small datasets using different strategies in the detection of proximal caries at different levels of severity on periapical radiographs.
GENERAL COMMENTS
The article is in-line with the journal topic, but some methodologic flaws should be improved. The investigation is interesting, and the present paper is recommended for publication to the present journal after major revision.
Introduction
The section is complete and well-written. More aspects according to the biofilms and bacteria etiology of dental caries should be investigated in the introduction part.
Materials and methods
The Cifar-10Net CNN network workflow process should be synthetized better through a scheme or figures.
The main flaw of the methodology process is the absence of standardization of the RXs technique and radiological dosage. Were they periapical scans? Bitewings?
In the figure 1 scheme the authors present very low-quality images with strong evidence of disparallelisms and artifacts. These aspects could represent a strong bias for interproximal caries detections in the clinical practice with a high rate of false-positive and false-negative rates.
Did you used a standardized thresholding gray scale for interproximal caries detection with Cifar-10Net CNN network? This aspect should be described accurately.
The authors provided only an inter-examiners concordance approach, but not a blinded intra-examiners assessment.
The calibration process should be described in this section.
Results
Line 140-141: “Consistency among examiners was checked before the revalidation, and a Kendall’s W coefficient of 0.830 (P < 0.001) showed strong consistency. Periapical radiographs were excluded when dispute arose.”
This part is appropriated in the results section
In my opinion, the interproximal enamel/dentin caries diagnosis should be accompanied by a clinical and radiographical/ evaluation. This study design should include a further clinical comparison group to indicate the false-positive and false-negative rates of the calibrated examiners and the Cifar-10Net CNN network process.
Discussions
The study null-hypothesis reject should be discussed. In accordance to the previous indications, the authors should include the aspects in the limits of the study findings.
The present investigation produced a clinical evaluation with no randomization/ blinding protocols with a high risk of bias.
The translational application of the study should be improved.
Author Response
1. The investigation is interesting, and the present paper is recommended for publication to the present journal after major revision.
Response: Thank you very much for your comments. These results were very encouraging.
Revised text: n.a.
2. Introduction. The section is complete and well-written. More aspects according to the biofilms and bacteria etiology of dental caries should be investigated in the introduction part.
Response: Thank you very much for your comments. The aspects according to the biofilms and bacterial etiology of dental caries have been added to the introduction part accordingly.
Revised text: “Dental caries occurs when plaque-associated bacteria produce acid that demineralizes the tooth. Controlling oral microbial biofilms is crucial for preventing dental caries. However, dental caries develops despite the use of antibiotics since the bacterial resistance occurs due to the excessive antibiotic use [5].” (Introduction section, Page 1, Lines 40-43).
3. Materials and methods. The Cifar-10Net CNN network workflow processshould be synthetizedbetter through a scheme or figures.
Response: Thank you very much for your comments. The workflow process of the Cifar-10Net CNN network has been synthetized through Figure 2 accordingly.
Revised text:
Figure 2. The workflow process of the CNN (Materials and Methods section, Page 6, Lines 218-219).
4. Materials and methods. The main flawof the methodology process is the absence of standardization of the RXs technique and radiological dosage. Were they periapical scans? Bitewings?
Response: Thank you very much for your comments. The study subjects in the present investigation were periapical scans, which were taken by radiologists. The paralleling technique was applied in the radiography process. Specifically, the X-ray tube head, digital sensor, and target tooth in the mandible (or maxilla) were aligned to allow radiographs to be exposed using the paralleling technique (Patel, S.; Wilson, R.; Dawood, A.; Mannocci, F. The detection of periapical pathosis using periapical radiography and cone beam computed tomography - part 1: pre-operative status. Int. Endod. J. 2012, 45, 702-710. DOI: 10.1111/j.1365-2591.2011.01989.x). However, the radiological dosage applied may change for individual oral conditions, for example, the soft tissue condition, which was consistent with a previous study (Shin, H.S.; Nam, K.C.; Park, H.; Choi, H.U.; Kim, H.Y.; Park, C.S. Effective doses from panoramic radiography and CBCT (cone beam CT) using dose area product (DAP) in dentistry. Dentomaxillofac. Radiol. 2014, 43, 20130439. DOI: 10.1259/dmfr.20130439). Nonetheless, the absence of standardization of the radiological dosage was regarded as a limitation of the present study. Details have been added to the revised text.
Revised text: “All the periapical radiographs were taken by radiologists applying the paralleling technique [34].” (Materials and Methods section, Page 3, Lines 133-134).
“First, radiological dosage standardization was lacking since the radiological dosage applied may change for individual oral conditions, for example, the soft tissue condition [51].” (Discussion section, Page 11, Lines 380-382).
5. Materials and methods. In the figure 1 scheme the authors present very low-quality images with strong evidence of disparallelisms and artifacts. These aspects could represent a strong bias for interproximal caries detections in the clinical practice with a high rate of false-positive and false-negative rates.
Response: Thank you very much for your comments. We apologize for the low quality of images in Figure 1. Accordingly, image quality has been enhanced to meet the journal standard. In addition, we agree with the reviewer that the accuracy of caries detection is closely related to image quality. To reduce the diagnostic bias that might be caused by image cropping, original periapical radiographs were also provided to the examiners for further needs. Details have been added to the revised text.
Revised text: “To reduce the diagnostic bias that might be caused by image cropping, original periapical radiographs were also provided to the 3 examiners for further needs.” (Materials and Methods section, Page 4, Lines 149-150).
Figure 1. Proximal caries detection on periapical radiographs using deep learning with different recognition modes. (Materials and Methods section, Page 5, Lines 210-212).
6. Materials and methods. Did you used a standardized thresholding gray scale for interproximal caries detection with Cifar-10Net CNN network? This aspect should be described accurately.
Response: Thank you very much for your question and comments. No standardized grayscale thresholding was used in the present CNN network because the Cifar-10Net CNN network is a nonlinear network instead of a regressor that needs a threshold (1. Fong, Y.; Huang, Y.; Gilbert, P.B.; Permar, S.R. chngpt: threshold regression model estimation and inference. BMC Bioinformatics. 2017, 18, 454; 2. Mathworks. Available online: https://www.mathworks.com/help/vision/examples/object-detection-using-deep-learning.html (accessed on 10 April 2020).). A classifier based on SoftMax was utilized to detect interproximal caries (Cho, C.; Choi, W.; Kim, T. Leveraging uncertainties in Softmax decision-making models for low-power IoT devices. Sensors (Basel). 2020, 20. DOI: 10.3390/s20164603). Nonetheless, the present CNN network, which is based on deep learning, may detect interproximal caries by synthetically distinguishing the grayscale change and edge morphology. Deep learning is characterized by automatically combining simple features to solve problems. Moreover, according to Lee et al., the caries detection system based on a convolutional neural network could learn the location and morphological alternation of caries lesions (Lee, J.H.; Kim, D.H.; Jeong, S.N.; Choi, S.H. Detection and diagnosis of dental caries using a deep learning-based convolutional neural network algorithm. J. Dent. 2018, 77, 106-111. DOI: 10.1016/j.jdent.2018.07.015). However, the learning process remains unknown. Details have been added to the revised text.
Revised text: “No standardized grayscale thresholding was used in the present CNN because the Cifar-10Net CNN is a nonlinear network instead of a regressor that needs a threshold [33, 41].” (Materials and Methods section, Page 4, Lines 194-196).
7. Materials and methods. The authors provided only an inter-examiners concordance approach, but not a blinded intra-examiners assessment.
Response: We apologize for the original description. Actually, the 3 examiners evaluated dental caries on periapical radiographs independently. For a single image, a consensus of the three examiners was required to determine the dental caries. Discussion was carried out when an inconsistent evaluation arose. Periapical radiographs were excluded when disputes remained unsolved. To reduce the diagnostic bias that might be caused by image cropping, original periapical radiographs were also provided to the 3 examiners for further needs. Details have been added to the revised text.
Revised text: “All images were clearly revalidated, and proximal caries (including enamel and dentin caries of permanent teeth) were distinguished from non-proximal caries by 3 endodontists independently. For a single image, a consensus of the 3 examiners was required to determine the dental caries. Discussion was carried out when inconsistent evaluations arose. Periapical radiographs were excluded when disputes remained unsolved. To reduce the diagnostic bias that might be caused by image cropping, original periapical radiographs were also provided to the 3 examiners for further needs.” (Materials and Methods section, Pages 3-4, Lines 142-144, 146-150)
8. Materials and methods.The calibrationprocess should be described in this section.
Response: Thank you very much for your comments. The dataset calibration process, called the normalization of images, has been described in the Materials and Methods accordingly. The normalization of images made it more convenient to accelerate the convergence of the neural network during the training process (Huang, L.; Qin, J.; Zhou, Y.; Zhu, F.; Liu, L.; Shao, L. Normalization techniques in training dnns: Methodology, analysis and application. arXiv 2020, arXiv:2009.12836.). The absence of standardization of the radiological dosage has been regarded as a limitation and has been added to the Discussion section.
Revised text: “Original datasets were then converted to grayscale images using uniform parameters, which was called the normalization of images.” (Materials and Methods section, Page 4, Lines 157-159).
First, radiological dosage standardization was lacking since changes in the applied radiological dosage may occur for individual oral conditions, for example, soft tissue conditions [51].” (Discussion section, Page 11, Lines 380-382).
9. Results. Line 140-141: “Consistency among examiners was checked before the revalidation, and a Kendall’s W coefficient of 0.830 (P < 0.001) showed strong consistency. Periapical radiographs were excluded when dispute arose.” This part is appropriated in the results section.
Response: Thank you for your comments. The part of “Consistency among examiners was checked before the revalidation, and a Kendall’s W coefficient of 0.830 (P < 0.001) showed strong consistency” has been moved to the Results section accordingly.
Revised text: “Consistency among examiners was checked before the revalidation, and Kendall’s W coefficient of 0.830 (P < 0.001) showed strong consistency.” (Results section, Page 6, Lines 234-235)
10. Materials and methods.In my opinion, the interproximal enamel/dentin caries diagnosis should be accompanied by a clinical and radiographical/ evaluation. This study design should include a further clinical comparison group to indicate the false-positive and false-negative rates of the calibrated examiners and the Cifar-10Net CNN network process.
Response: Thank you very much for your comments. We agree with the reviewer that the interproximal enamel/dentin caries diagnosis should be accompanied by a clinical and radiographical evaluation. According to previous studies, in the absence of a “hard” reference test, evaluating interproximal caries by three endodontists was referenced as the “gold standard” (1. Cantu, A.G.; Gehrung, S.; Krois, J.; Chaurasia, A.; Rossi, J.G.; Gaudin, R.; Elhennawy, K.; Schwendicke, F. Detecting caries lesions of different radiographic extension on bitewings using deep learning. J. Dent. 2020, 100, 103425. DOI: 10.1016/j.jdent.2020.103425; 2. Lee, J.H.; Kim, D.H.; Jeong, S.N.; Choi, S.H. Detection and diagnosis of dental caries using a deep learning-based convolutional neural network algorithm. J. Dent. 2018, 77, 106-111. DOI: 10.1016/j.jdent.2018.07.015). However, it was a limitation that only the radiographical evaluation has been conducted, lacking a clinical evaluation. A further clinical comparison group to indicate the false-positive and false-negative rates of the calibrated examiners and the Cifar-10Net CNN process will be developed in further research. Details have been added to the Discussion section.
Revised text: “Second, in the absence of a “hard” reference test, only radiographical evaluations were conducted, lacking clinical evaluations [9,35]. Last, a further clinical comparison group (such as combining the clinical records when caries evaluations are conducted) to indicate the false-positive and false-negative rates of the calibrated examiners and the Cifar-10Net CNN process could be considered to further verify the current findings.” (Discussion section, Page 11, Lines 385-386, 396-399).
11. Discussions.The study null-hypothesis reject should be discussed. In accordance to the previous indications, the authors should include the aspects in the limits of the study findings.
Response: Thank you very much for your comments. The rejection of the null hypothesis for the study has been added to the Discussion section and included in the limitations accordingly.
Revised text: “In particular, the CNN trained with EE and IR strategies performed better than that with the IS strategy, and the CNN trained with the EE strategy achieved higher accuracy and sensitivity than human eyes in the detection of proximal caries.” (Discussion section, Page 9, Lines 277-280).
“Furthermore, inconsistent with the CNN trained with EE and IR strategies that performed well, the CNN trained with the IS strategy, namely, the IS recognition mode, behaved indiscriminately in detecting proximal caries and showed a poorer performance than that achieved by human eyes.” (Discussion section, Page 11, Lines 386-389).
12. Discussions.The present investigation produced a clinical evaluation with no randomization/ blinding protocols with a high risk of bias.
Response: Thank you very much for your comments. We apologize for the lack of details about the randomization/blinding protocols in the present clinical evaluation. Actually, the periapical radiographs were collected following the randomization principle. The three examiners evaluated dental caries on periapical radiographs independently and were blinded to the subject group to which they belonged. Details have been added to the Materials and Methods section.
Revised text: “Anonymous periapical radiographs were collected from patients who visited the Hospital of Stomatology, Fujian Medical University, from 2019 to 2020, following the randomization principle. All images were clearly revalidated, and proximal caries (including enamel and dentin caries of permanent teeth) were distinguished from non-proximal caries by 3 endodontists independently. For a single image, a consensus of the 3 examiners was required to determine the dental caries. Discussion was carried out when inconsistent evaluations arose. Periapical radiographs were excluded when disputes remained unsolved. To reduce the diagnostic bias that might be caused by image cropping, original periapical radiographs were also provided to the 3 examiners for further needs. All 800 periapical radiographs were given a random number by using the RAND function and were randomly assigned to the training or test dataset by using the data sorting function in Microsoft Excel (Microsoft office 2016, Microsoft, USA). Subsequently, a training and validation dataset (n = 600) and a test dataset (n = 200) were randomly generated.” (Materials and Methods section
, Pages 3-4, Lines 131-133, 142-144, 146-150, 153-157).
13. Discussions.The translational application of the study should be improved.
Response: Thank you very much for your comments. The translational application of the study would be improved through the following enhancements: First, recording the applied dosage when taking periapical radiographs can be utilized to obtain standard images, which should be done in cooperation with radiologists. Second, clinical records can be combined when caries evaluations are conducted; thus, a clinical comparison group could be employed to indicate the false-positive and false-negative rates of the calibrated examiners and the Cifar-10Net CNN process. Furthermore, a larger sample size and balanced dataset (different levels of caries severity) could be considered to enhance the generalizability of the present approach and to explore the impact of using the network on treatment decisions. Details have been added to the Discussion section.
Revised text: “First, radiological dosage standardization was lacking since changes in the applied radiological dosage may occur for individual oral conditions, for example, soft tissue conditions [51]. Establishing a record of the applied dosage when taking periapical radiographs could be considered for obtaining standard images, which should be accomplished in cooperation with radiologists. Second, in the absence of a “hard” reference test, only radiographical evaluations were conducted, lacking clinical evaluations [9, 35]. Furthermore, inconsistent with the CNN trained with EE and IR strategies that performed well, the CNN trained with the IS strategy, namely, the IS recognition mode, behaved indiscriminately in detecting proximal caries and showed a poorer performance than that achieved by human eyes. A larger sample size and balanced dataset (different levels of caries severity) could be considered to enhance the generalizability of the present approach and explore the impact of using the network on treatment decisions. Last, a further clinical comparison group (such as combining the clinical records when caries evaluations are conducted) to indicate the false-positive and false-negative rates of the calibrated examiners and the Cifar-10Net CNN process could be considered to further verify the current findings.” (Discussion section, Page 11, Lines 380-389, 391-393, 396-399).

Reviewer 2 Report
- There is a certain inconsistency at work. In lines 243-246 the authors write that the proposed method is partially accepted. But in 350-351 "performed excellently in detecting", then the method worked or not ?!
- The improvement of the method of early caries detection proposed by the authors is a very desirable phenomenon.
- The description of the method described by the authors is not precise and raises a number of reservations. For example, to verify the correctness of the proposed method, a team of human experts was used, consisting of three people with min. 5 years of experience. The authors do not provide any results of expert assessments.
Author Response
1. Discussion.There is a certain inconsistency at work. In lines 243-246 the authors write that the proposed method is partially accepted. But in 350-351 "performed excellently in detecting", then the method worked or not ?!
Response: Thank you very much for your comments. We apologize for that the original description caused confusion. The method of training the CNN with the EE strategy definitely worked better than human eyes. Thus, the conclusion that the deep learning-based CNN trained with the EE strategy performed excellently in detecting proximal caries on periapical radiographs was drawn. However, only the CNN trained with the EE strategy, instead of all three training strategies, was more sufficient and accurate than human eyes in detecting proximal caries. Therefore, the hypothesis that the CNN trained with different strategies would be more sufficient and accurate than human eyes in detecting proximal caries was partially accepted. Details have been added to the Discussion section.
Revised text: “In particular, the CNN trained with EE and IR strategies performed better than that with the IS strategy, and the CNN trained with the EE strategy achieved higher accuracy and sensitivity than human eyes in the detection of proximal caries.” (Discussion section, Page 9, Lines 277-280).
2. Discussion. The improvement of the method of early caries detection proposed by the authors is a very desirable phenomenon.
Response: Thank you very much for your comments, and we hope that the elaboration of the following aspects, including the improvement of the method, could obtain agreement from you. (1) Since early intervention can block or reverse the process of dental caries, finding an approach to detect initial caries, especially proximal caries, efficiently is of great significance (1. Cantu, A.G.; Gehrung, S.; Krois, J.; Chaurasia, A.; Rossi, J.G.; Gaudin, R.; Elhennawy, K.; Schwendicke, F. Detecting caries lesions of different radiographic extension on bitewings using deep learning. J. Dent. 2020, 100, 103425. DOI: 10.1016/j.jdent.2020.103425; 2. Gomez, J.; Tellez, M.; Pretty, I.A.; Ellwood, R.P.; Ismail, A.I. Non-cavitated carious lesions detection methods: A systematic review. Community Dent. Oral Epidemiol. 2013, 41, 54-66. DOI: 10.1111/cdoe.12021). The proposed method should be regarded as a computer-aided caries detection system in clinical practice, in which the system may be more capable of detecting initial caries on radiographs than human eyes. Certainly, combining clinical evaluations should not be discarded. (2) Standardization of the radiological dosage was absent in the present study; thus, recording the application dosage when taking periapical radiographs could be considered for obtaining standard images, which should be done in cooperation with radiologists. (3) Pertinently, the generalizability of the present network and the impact of using it on treatment decisions should be further explored with a larger sample size and balanced dataset (different levels of caries severity). (4) Additionally, we sincerely acknowledge that it was a limitation that only the radiographical evaluation was conducted, lacking a clinical evaluation, in the process of the interproximal enamel/dentin caries being evaluated by 3 examiners. According to previous studies, in the absence of a “hard” reference test, evaluating interproximal caries by three endodontists was referenced as the “gold standard” (1. Cantu, A.G.; Gehrung, S.; Krois, J.; Chaurasia, A.; Rossi, J.G.; Gaudin, R.; Elhennawy, K.; Schwendicke, F. Detecting caries lesions of different radiographic extension on bitewings using deep learning. J. Dent. 2020, 100, 103425. DOI: 10.1016/j.jdent.2020.103425; 2. Lee, J.H.; Kim, D.H.; Jeong, S.N.; Choi, S.H. Detection and diagnosis of dental caries using a deep learning-based convolutional neural network algorithm. J. Dent. 2018, 77, 106-111. DOI: 10.1016/j.jdent.2018.07.015). A further clinical comparison group, such as combining the clinical records when caries evaluations are conducted, to indicate the false-positive and false-negative rates of the calibrated examiners and the Cifar-10Net CNN network process would be developed in the future research. Details have been added to the Discussion section.
Revised text: “First, radiological dosage standardization was lacking since changes in the applied radiological dosage may occur for individual oral conditions, for example, soft tissue conditions [51]. Establishing a record of the applied dosage when taking periapical radiographs could be considered for obtaining standard images, which should be accomplished in cooperation with radiologists. Second, in the absence of a “hard” reference test, only radiographical evaluations were conducted, lacking clinical evaluations [9, 35]. Furthermore, inconsistent with the CNN trained with EE and IR strategies that performed well, the CNN trained with the IS strategy, namely, the IS recognition mode, behaved indiscriminately in detecting proximal caries and showed a poorer performance than that achieved by human eyes. Moreover, the sample size was unbalanced at different levels of caries severity, which may have impacted the present findings. A larger sample size and balanced dataset (different levels of caries severity) could be considered to enhance the generalizability of the present approach and to explore the impact of using the network on treatment decisions. Additionally, the recognition rate was partly sacrificed to increase the training and recognition speed [38]. The number of network layers should be increased to improve the recognition rate in future studies. Last, a further clinical comparison group (such as combining the clinical records when caries evaluations are conducted) to indicate the false-positive and false-negative rates of the calibrated examiners and the Cifar-10Net CNN process could be considered to further verify the current findings.” (Discussion section, Page 11, Lines 380-399).
3. Results. The description of the method described by the authors is not precise and raises a number of reservations. For example, to verify the correctness of the proposed method, a team of human experts was used, consisting of three people with min. 5 years of experience. The authors do not provide any results of expert assessments.
Response: Thank you very much for your comments. We apologize for that the original writing caused a misunderstanding. In fact, the results of expert assessments, demonstrated as a reference dataset, are exhibited in Table 1. As mentioned in the text, all 800 periapical radiographs included were randomly divided into a training and validation dataset (n = 600) and a test dataset (n = 200). Caries lesions of the dataset were marked using a training image labeller (TIL) based on the agreement among the 3 examiners and shown as the region of interest (ROI). According to the ROI, the level of caries severity was then evaluated and considered as the reference (the “gold standard”). Details have been added to the Results section.
Revised text: “The caries occurrences in proximal surfaces in the reference dataset, that is, the evaluation from the 3 examiners, are depicted in Table 1.” (Results section, Page 6, Lines 235-237).

Reviewer 3 Report
The paper needs some major and minor revisions, as follows:
- There are some issues related to the novelty, the authors did not clarified the main differences between this approach and existing ones. Please clarify those differences to convey the reader about the main contribution of our paper.
- Some figures quality must be enhance.
- More details about the approach is needed. for example, add pseudocodes of applied methods to make it easier to the readers.
- Explain the parameter settings of all tested methods. How did you guarantee fair comparisons?
- The conclusion should also discuss the challenges, limitations, and future directions.
Author Response
The paper needs some major and minor revisions, as follows:
1. Discussion.There are some issues related to the novelty, the authors did not clarified the main differences between this approach and existing ones. Please clarify those differences to convey the reader about the main contribution of our paper.
Response: Thank you very much for your comments. The main differences between this approaches and existing approaches were clarified in Discussion accordingly.
Revised text: “Compared to the approaches utilizing GoogLeNet and U-Net, different models equipped with different loss functions and combinations of the parameters contributed to the main differences among existing approaches and the present proposal [9, 35]. Therefore, a preprocessing strategy that is common but well suited to medical images, such as the EE strategy, was proposed to preprocess periapical radiographs that are commonly used in clinical practice [45].” (Discussion section, Page 10, Lines 360-363, 371-373).
2. Materials and methods & Results.Some figures quality must be enhanced.
Response: Thank you very much for your comments. The quality of all figures has been enhanced to meet the journal standard. The revised figures have been inserted in the text, for example, Figure 1 in the Materials and Methods section and Figure 3 in the Results section.
Revised figures:
Figure 1. Proximal caries detection on periapical radiographs using deep learning with different recognition modes. (Materials and Methods section, Page 5, Lines 210-212).
Figure 3. The ROC curves of different recognition modes and human eyes. (Results section, Page 7, Lines 255-256).
3. Materials and methods.More details about the approach is needed. for example, add pseudocodes of applied methods to make it easier to the readers.
Response: Thank you very much for your comments. To make it easier to the readers, the main functions of relevant codes and parameters of proposed methods (including data processing and the training process) have been added to the footnote in Figure 1. The training model was explained by a more detailed flowchart (Figure 2). In addition, the complete function codes of the proposed methods would be provided as an Appendix if needed. More details have been added to the Materials and Methods section.
Revised text and figure: “The main functions of relevant codes and some parameters in Data processing and Training were conducted as follows: IS, function imgf = fenge(rgb); EE, function imgCanny = edge_canny(I,gaussDim,sigma,percentOfPixelsNotEdges,thresholdRatio); Image superposition, function C = diejia(pic_1,pic_2); Training, function training = trainRCNNObjectDetector (Un-named, mylayers, options, ... 'NegativeOverlapRange', [0 0.3]). (Materials and Methods section, Page 5, Lines 213-217).
Figure 2. The workflow process of the CNN. (Materials and Methods section, Page 6, Lines 218-219)
4. Materials and methods.Explain the parameter settings of all tested methods. How did you guarantee fair comparisons?
Response: Thank you very much for your comments. The neural network model was chosen based on previous studies. Then, the parameter settings (including the base learning rate, loss value, and epochs) were determined based on a grid search and modified according to the loss value. For example, the base learning rate was selected and adjusted according to the loss change. The epochs were adjusted according to the change of loss to avoid overfitting. To guarantee fair comparisons, the metric of the loss value was employed as the only standard and parameter variables were controlled. More details have been added to the Materials and Methods section.
Revised text: “Taking the loss value as the evaluation metric, a base learning rate of 0.0001 was set, and 400 epochs were run.” (Materials and Methods section, Page 4, Lines 192-193).
5. Conclusions.The conclusion should also discuss the challenges, limitations, and future directions.
Response: Thank you very much for your comments. The challenges, limitations, and future directions have been added to the Conclusions section accordingly.
Revised text: “Within the limitations of a lack of standardization of the radiological dosage and a small sample size, we prudently concluded that the deep learning-based CNN trained with the EE strategy performed excellently in detecting proximal caries on periapical radiographs; different training strategies, such as image preprocessing, could be considered to improve the accuracy of the CNN model, especially when a small dataset was used. Pertinently, the present proposed method proposed should be regarded as a computer-aided caries detection system in clinical practice, in which clinical evaluations should be combined and not discarded. However, the challenges of how the proposed method could be generalized and applied to treatment decisions should be considered. Additionally, regarding the limitation of only conducting radiographical evaluations, a further clinical comparison group to indicate the false-positive and false-negative rates of the calibrated examiners and the Cifar-10Net CNN process could be considered to further verify the current findings.” (Conclusions section, Page 11, Lines 401-413).

Reviewer 4 Report
Thanks for recommending me as a reviewer. In this paper, authors aimed to evaluate the performance of convolutional neural networks (CNNs) that were trained with small datasets using different strategies in the detection of proximal caries at different levels of severity on periapical radiographs. Small datasets containing 800 periapical radiographs were randomly categorized into a training and validation dataset (n = 600) and a test dataset (n = 200). A pretrained Cifar-10Net CNN network was used in the present study. If authors complete minor revisions, the quality of the study will be further improved.
- The introduction section is well written.
- Abbreviations used in the legends of Figures 2 and 3 should include their full names in footnotes.
- The discussion section is well written. On the other hand, the conclusion section is sparse. If authors write a richer conclusion section, it can help readers understand.
Author Response
Reviewer 4
Thanks for recommending me as a reviewer. If authors complete minor revisions, the quality of the study will be further improved.
1. Introduction.The introduction section is well written.
Response: Thank you very much for your comments.
Revised text: n.a.
2. Results. Abbreviations used in the legends of Figures 2 and 3 should include their full names in footnotes.
Response: Thank you very much for your comments. Full names of the abbreviations used in the legends of Figures 2 and 3 (Figures 3 and 4 in the revised text) have been added in the footnotes accordingly.
Revised text: ROC = receiver operating characteristic; IR = image recognition; EE = edge extraction; IS = image segmentation; P-R = precision-recall (Results section, Pages 7-8, Lines 257-258, 261).
3. Discussion & Conclusions.The discussion section is well written. On the other hand, the conclusion section is sparse. If authors write a richer conclusion section, it can help readers understand.
Response: Thank you very much for your comments. Richer conclusions, including the challenges, limitations and further directions, have been elaborated in the Conclusions section accordingly.
Revised text: “Within the limitations of a lack of standardization of the radiological dosage and a small sample size, we prudently concluded that the deep learning-based CNN trained with the EE strategy performed excellently in detecting proximal caries on periapical radiographs; different training strategies, such as image preprocessing, could be considered to improve the accuracy of the CNN model, especially when a small dataset was used. Pertinently, the present proposed method proposed should be regarded as a computer-aided caries detection system in clinical practice, in which clinical evaluations should be combined and not discarded. However, the challenges of how the proposed method could be generalized and applied to treatment decisions should be considered. Additionally, regarding the limitation of only conducting radiographical evaluations, a further clinical comparison group to indicate the false-positive and false-negative rates of the calibrated examiners and the Cifar-10Net CNN process could be considered to further verify the current findings.” (Conclusions section, Page 11, Lines 401-413).

Round 2
Reviewer 1 Report
The paper has been improved
Reviewer 2 Report
Thank you for the explanations sent. The work in this form can be published
Reviewer 3 Report
The authors addressed all comments raised in the previous round.
This version is ok, and can be accepted.